# IMPROVING SUBGRAPH REPRESENTATION LEARNING VIA MULTI-VIEW AUGMENTATION

## ABSTRACT

Subgraph representation learning based on Graph Neural Network (GNN) has exhibited broad applications in scientific advancements, such as predictions of molecular structure–property relationships and collective cellular function. In particular, graph augmentation techniques have shown promising results in improving graph-based and node-based classification tasks. Still, they have rarely been explored in the existing GNN-based subgraph representation learning studies. In this study, we develop a novel multi-view augmentation mechanism to improve subgraph representation learning models and thus the accuracy of downstream prediction tasks. Our augmentation technique creates multiple variants of subgraphs and embeds these variants into the original graph to achieve highly improved training efficiency, scalability, and accuracy. Benchmark experiments on several real-world biological and physiological datasets demonstrate the superiority of our proposed multi-view augmentation techniques in subgraph representation learning.

## 1 INTRODUCTION

Subgraph representation learning using Graph Neural Networks (GNNs) can be broadly applied to various subgraph-related tasks in many fields of science and technology. As an outstanding example, the PPI (Protein–Protein Interaction) network (Zitnik et al., 2018) uses nodes, edges, and subgraphs to represent single proteins, their interactions, and the set of interacting proteins, respectively. GNNs can be used to predict the biological processes (PPI-BP), cell component (PPI-CC), and molecular function (PPI-MF) by classifying the functionality of a subgraph (i.e., a group of proteins) in the PPI network. Another example is to apply GNN to fragment-based quantum chemical theory where each fragment in a crystal or aggregate is a subgraph and subgraph representation learning can predict the quantitative interactions between different fragments. Although applying GNNs to subgraph-related tasks (Alsentzer et al., 2020; Kim & Oh, 2022; Wang & Zhang, 2021) starts to draw some attention, none of them have implemented graph augmentation techniques to improve task accuracy.

This work presents a novel multi-view approach to augment graphs for improving accuracy of subgraph classification tasks. Inspired by the effectiveness of graph contrastive learning Hassani & Khasahmadi (2020); Zhu et al. (2020); You et al. (2020), our basic idea is to create multiple views of a subgraph by augmenting it, learn the embedding for each of the view, and then combine the representations for predicting the label of the subgraph. The rationale behind it is that the augmented subgraphs (i.e., the multiple views) essentially form an ensemble, which could provide more robust signal in determining the properties of the subgraph.

The basic idea poses a fundamental challenge in how to efficiently create augmented subgraphs. Augmenting the entire graph to produce different views of the same subgraph is not scalable because the size of the augmented graph will grow linearly with the number of views. Figure 1(c) illustrates the problem. With only one additional view, GNNs need to conduct forward and backward propagations on two independent graphs (i.e., the original graph and the augmented graph) during training, doubling the training cost. We address the efficiency issue by embedding augmented subgraphs in the original graph, significantly decreasing the demand for GPU resources. In this case, the computation of the embeddings for the augmented subgraphs can share intermediate representations within their neighborhood. Figure 1(d) illustrates an alternative efficient design where the augmented subgraphs are embedded into an augmented graph, instead of the original graph. We empirically validate

that preserving the original view of subgraphs is essential for multi-view augmentation to improve task accuracy.

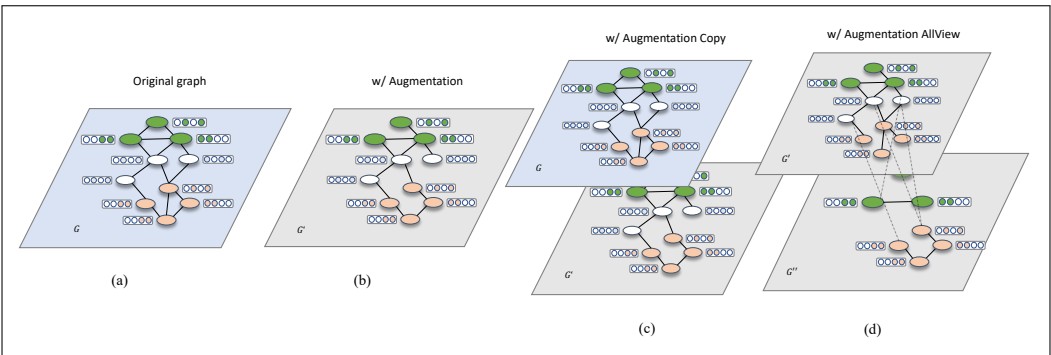

Figure 1: Illustration of graph augmentation approaches. (a) The original graph $\mathcal{G}$ contains two subgraphs (colored in orange and blue). (b) Augmented subgraphs are created by randomly dropping some edges in $\mathcal{G}$. The new graph $\mathcal{G}'$ is called the augmented graph. (c) A graph with two independent components where $\mathcal{G}$ is the original graph and $\mathcal{G}'$ is the augmented graph. Learning on this graph doubles the training cost. (d) Augmented subgraphs $\mathcal{G}''$ are embedded into an augmented graph $\mathcal{G}'$.

In summary, this work makes the following contributions:

- This work proposes a novel multi-view augmentation strategy to improve the accuracy of subgraph-based learning tasks. This study is the first to explore the benefits of graph augmentation techniques in subgraph representation learning.
- The proposed multi-view augmentation strategy dynamically binds augmented subgraph views to the whole graph to drop exaggerated GPU resource consumption in order to achieve highly-improved training efficiency and task accuracy.
- Empirical evaluations on three subgraph datasets demonstrate that our augmentation approach can improve existing subgraph representation learning by 0.3%–2.9% in accuracy, which is on average 1.1% higher than general graph augmentation techniques DropEdge and GAug-M.

## 2 RELATED WORKS

**Subgraph Representation Learning** Subgraph representation learning using GNNs has gained substantial attention these years (Meng et al., 2018) due to its broad applications in scientific domains. Outstanding examples include SubGNN (SubGraph Neural Network) (Alsentzer et al., 2020), which routes messages for internal and border properties within sub-channels of each channel, including neighborhood, structure, and position. After that, the anchor patch is sampled and the features of the anchor patch are aggregated to the connected components of the subgraph through six sub-channels. GLASS (Wang & Zhang, 2021) employs a labeling trick (Zhang et al., 2021) and labels nodes belonging to any subgraph to boost plain GNNs on subgraph tasks. S2N (Subgraph-To-Node) (Kim & Oh, 2022) translates subgraphs into nodes and thus reduces the scale of the input graph. These approaches focus on developing novel subgraph-based GNNs to improve task accuracy, but they have never implemented graph augmentation techniques.

**Graph Augmentation** Data augmentation is a vital part of deep learning. Many general graph augmentation techniques have been proposed to improve task accuracy recently. For node classification tasks, Rong et al. (2020) proposes DropEdge to randomly drop the edges in a graph to enlarge the support of the training distribution. DGI (Deep Graph Infomax) (Veličković et al., 2019) perturbs the nodes by performing a row-wise swap of the input feature matrix while the adjacency matrix remains unchanged, generating negative samples for comparison learning and maximizing the mutual information of input and output. GAug (Zhao et al., 2021) generates and removes edges of the graph by training an edge predictor to finally achieve the effect of high connectivity between nodes

within the same class and low connectivity between nodes from different classes. NeuralSparse (Neural Sparsification) (Zheng et al., 2020) proposes a supervised graph sparsification technique that improves generalization by learning to remove potentially task-irrelevant edges from the input graph. GraphCL (You et al., 2020) points out that different data augmentation techniques introduce different advantages in graph learning tasks in different domains. For example, edge perturbation can enhance learning in social network graphs, but can be counterproductive in compound graphs learning by destroying the original information. SUBG-CON (SUBGraph CONtrast) (Jiao et al., 2020) samples a series of subgraphs containing regional neighbors from the original graph as training data to serve as an augmented node representation. Although these methods show promising results for augmenting graphs for node- and graph-based downstream tasks, they are not designed for augmenting subgraphs for subgraph-based tasks.

**Multi-view Graph Learning**   Multi-view representation learning on graphs has attracted significant attention because they capture different properties on the same graph. Hassani et al. (Hassani & Khasahmadi, 2020) introduce a multi-view graph learning manner to perform contrastive learning. O2MAC (One2Multi graph AutoenCoder) (Fan et al., 2020) proposes a multi-view-based autoencoder to promote self-supervised learning. MV-GNN (Multi-View Graph Neural Network) (Ma et al., 2020) utilizes two MPNNs (Message Passing Neural Networks) (Gilmer et al., 2017) to encode atom and bond information respectively via multi-view graph construction. They construct multi-view graphs to express different levels of information in a graph, which is an intuitive and efficient way of building augmented graphs. Our work also leverages multi-view–based augmentation but focuses on subgraph-based tasks.

## 3   NOTATIONS AND PRELIMINARIES

### 3.1   NOTATIONS

Let $\mathcal{G} = (\mathbb{V}, \mathbb{E}, \boldsymbol{X})$ denote a graph, where $\mathbb{V} = \{1, 2, .., N\}$ represents the node set, $\mathbb{E} \subseteq \mathbb{V} \times \mathbb{V}$ represents the edge sets, and $\boldsymbol{X}$ is the matrix that represents the corresponding node feature. $X_i$, the $i^{\text{th}}$ row of $\boldsymbol{X}$, represents the features associated with the $i^{\text{th}}$ node. Let $v_i$ denote a node in $\mathcal{G}$. The adjacency matrix $\boldsymbol{A} \in \{0, 1\}^{N \times N}$, where $a_{ij} = 1$ denotes that $(v_i, v_j) \in \mathbb{E}$. $\mathcal{G}_S = (\mathbb{V}_S, \mathbb{E}_S, \boldsymbol{X}_S)$ denotes a subgraph of $\mathcal{G}$, where $\mathbb{V}_S \subseteq \mathbb{V}$, $\mathbb{E}_S \subseteq \mathbb{E} \cap (\mathbb{V}_S \times \mathbb{V})_S$, and $\boldsymbol{X}_S$ stacks the rows of $\boldsymbol{X}$ belonging to $\mathbb{V}_S$. The adjacency matrix of a subgraph $\mathcal{G}_S$ is $\boldsymbol{A}_S$.

### 3.2   SUBGRAPH REPRESENTATION LEARNING

Given the set of subgraphs $\mathbb{S} = \{\mathcal{G}_{S_1}, \mathcal{G}_{S_2}, .., \mathcal{G}_{S_n}\}$ and their labels $\mathbb{T} = \{t_{S_1}, t_{S_2}, ..., t_{S_n}\}$, the goal of *Subgraph Representation Learning* is to learn a representation embedding $h_{S_i}$ for each subgraph $\mathcal{G}_{S_i}$ to predict the corresponding $t_{S_i}$.

### 3.3   GRAPH AUGMENTATION

In the present work, we illustrate our multi-view augmentation scheme based on two typical existing graph augmentation strategies, DropEdge (Rong et al., 2020) and GAug-M (Zhao et al., 2021).

DropEdge is a graph data perturbation strategy that randomly drops edges in a graph (Rong et al., 2020) so that it enlarges the training support to improve the performance of GNNs on node-level tasks. We employ DropEdge for each subgraph to generate an augmented subgraph, by generating a stochastic boolean mask $\boldsymbol{M}_p \in \mathbb{R}^{m \times m}$, where $m$ is the number of nodes in the subgraph and $p$ represents the rate of dropping edges. The new adjacency matrix becomes $\boldsymbol{A}'_S = \boldsymbol{A}_S - \boldsymbol{M}_p \odot \boldsymbol{A}_S$, where $\odot$ means the element-wise product.

GAug-M (Zhao et al., 2021) is a graph data augmentation strategy that leverages neural edge predictors to promote intra-class and demote inter-class edges so as to form new edge weights. It contains a two-stage training schema. In the first step, we use VGAE (Variational Graph Auto-Encoders) (Kipf & Welling, 2016) as the edge predictor to get an edge-probability matrix $\boldsymbol{M}$, which describes the graph's probabilistic connectivity, $\boldsymbol{M} = \sigma(\boldsymbol{Z}\boldsymbol{Z}^T)$, where $\boldsymbol{Z} = \text{GNN}(\boldsymbol{A}, \boldsymbol{X})$). Denote $|\mathbb{E}|$ as the number of edges in graph $\mathcal{G}$. In the second step, we use the probability matrix $\boldsymbol{M}$, to make the top $i|\mathbb{E}|$ non-edges with the highest edge probabilities to be connected, and the least $j|\mathbb{E}|$ edges with the

lowest edge probabilities to be disconnected to produce an augmented graph from $\mathcal{G}$ to $\mathcal{G}'$, where $i, j \in \{0, 1\}$.

# 4 METHODS

In this section, we present our proposed multi-view augmentation approach, following by analyses of the computational complexity and discussions on the shortage of the alternatives shown in Figure 1 and the advance of our multi-view approach.

## 4.1 METHODOLOGY

Figure 2 illustrates the basic idea of the multi-view augmentation that is implemented in the present study. At each forward step, we generate augmented views of subgraphs in this batch by randomly perturbing original subgraphs with a particular graph data augmentation strategy. After that, we add the augmented subgraphs to the original graph and feed the new graph into a subgraph-specific neural network. Here, we obtain subgraph embeddings of both the original subgraph and the augmented subgraph. These embeddings are fed into a pooling function to generate a single subgraph embedding for each subgraph, which is used for downstream subgraph-based tasks. Meanwhile, to maximize the agreement between the original subgraph embeddings and the augmented subgraph embeddings, we utilize the contrastive loss between the original graph and augmented graphs.

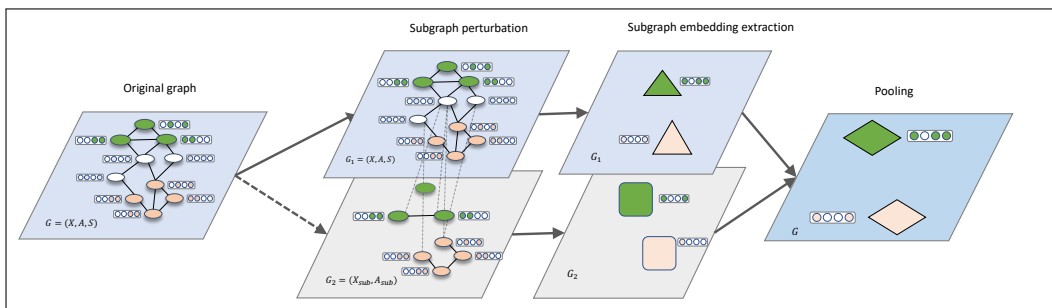

Figure 2: Overview of our proposed subgraph augmentation approach. The two subgraphs in the original graph are colored in green and orange. We first generate multi-subgraph views via stochastic augmentation. Following that we connect the augmented subgraph to the remaining part of the original graph, by adding edges that link the augmented subgraph and the whole graph. After feeding forward the whole graph into subgraph-specific GNNs, we extract the subgraph embeddings of different views, respectively (triangles and squares). Ultimately, we fuse the embeddings of different views by a pooling function and obtain the augmented subgraph embeddings (diamonds).

After subgraph augmentation, we obtain augmented subgraphs $\mathcal{G}'_S = (\mathbb{V}'_S, \mathbb{E}'_S, \boldsymbol{X}'_S)$. We enrich the original graph to include both the augmented subgraph and the original subgraph. The enriched graph is thus called a *Multi-View Graph*. Mathematically, the multi-view graph $\mathcal{G}' = (\mathbb{V}', \mathbb{E}', \boldsymbol{X}')$ where $\mathbb{V}' = \mathbb{V} \cup \mathbb{V}'_S$. The consequent adjacency matrix becomes

$$\boldsymbol{A}' = \begin{bmatrix} \boldsymbol{A} & \boldsymbol{A}[:, \mathbb{V}'_S] \\ \boldsymbol{A}[\mathbb{V}'_S, :] & \boldsymbol{A}_S \end{bmatrix}. \tag{1}$$

Feeding forward the multi-view graph into subgraph-specific neural networks, by selecting the subgraph and augmented subgraph nodes by masks $M_{S_O}, M_{S_A}$, we can get the embeddings of both the augmented subgraph and the original subgraph and denote them as $h_{S_O}$ and $h_{S_A}$, respectively. We fuse different subgraph embeddings into one embedding by applying a pooling function (e.g., MaxPool or AvgPool):

$$\mathbb{H} = \text{GNN}(\mathcal{G}') \tag{2}$$

$$h_{S_O}, h_{S_A} = \mathbb{H}[\boldsymbol{M}_{S_O}], \mathbb{H}[\boldsymbol{M}_{S_A}], \tag{3}$$

$$h_S = \text{Pool}(h_{S_O}, h_{S_A}) \tag{4}$$

With the learned subgraph embeddings, we can predict the subgraph properties by applying a MLP (Multi-Layer Perception), (Hastie et al., 2009)

$$\hat{t}_S = \text{softmax}(\text{MLP}\,(h_S)). \tag{5}$$

Meanwhile, we implement the contrastive loss between the augmented views, which enforces the embeddings of the original subgraph and the generated subgraph embedding to be close and those of augmented subgraphs not generated by this subgraph to be distant. Let $\lambda$ be the coefficient to control the contrastive loss contribution, the total loss function becomes

$$\mathcal{L} = \mathcal{L}_{\text{cls}}(t_S, \hat{t_S}) + \lambda \mathcal{L}_{\text{contrast}}(h_{S_O}, h_{S_A}). \tag{6}$$

## 4.2 COMPUTATIONAL COMPLEXITY ANALYSIS

The proposed subgraph multi-view augmentation is independent to augmentation strategies or subgraph-specific GNNs. Thus, we can train the model in an end-to-end fashion, which means there's neither modification at the beginning of each epoch nor after the forward. We can simply analyze the additional computational complexity made by the augmentation, the corresponding augmented graph inference at each forward step.

Let the number of nodes in the original graph be $|\mathbb{V}|$, the number of edges be $|\mathbb{E}|$, the number of the $i^{\text{th}}$ subgraph node be $|\mathbb{V}_{S_i}|$, the number of the $i^{\text{th}}$ subgraph node be $|\mathbb{E}_{S_i}|$, and the training setting for batch size be $b$. Therefore, the expectation of $|\mathbb{V}_S|$, $\mathbb{E}[|\mathbb{V}_S|]$ is going to be $\mathbb{E}[|\mathbb{V}_S|] = \dfrac{\sum_i |\mathbb{V}_{S_i}|}{b}$. Likewise, the expectation of $|\mathbb{E}_S|$, $\mathbb{E}[|\mathbb{E}_S|]$ is going to be $\mathbb{E}[|\mathbb{E}_S|] = \dfrac{\sum_i |\mathbb{E}_{S_i}|}{b}$.

Considering a graph neural network for which the computational complexity is $O(f_{\text{GNN}}(|\mathbb{V}|, |\mathbb{E}|))$, and the memory complexity is $O(g_{\text{GNN}}(|\mathbb{V}|, |\mathbb{E}|))$, the computational complexity for an augmentation strategy is $O(f_{\text{Aug}}(|\mathbb{V}|, |\mathbb{E}|))$, and the memory complexity is $O(g_{\text{Aug}}(|\mathbb{V}|, |\mathbb{E}|))$. Because we augment on subgraphs only during the augmentation stage, the computational complexity is $O(f_{\text{Aug}}(\mathbb{E}[|\mathbb{V}_S|], |\mathbb{E}[|\mathbb{E}_S|]))$, and the memory complexity is $O(g_{\text{Aug}}(\mathbb{E}[|\mathbb{V}_S|], \mathbb{E}[|\mathbb{E}_S|]))$ With regard to the forward step, the total computational complexity of the whole graph is $|\mathbb{V}| + \mathbb{E}[|\mathbb{V}_S|]$ and that of edges is $|\mathbb{E}| + \mathbb{E}[|\mathbb{E}_S|]$. In this way, the inference computational complexity is $O(f_{\text{GNN}}(|\mathbb{E}| + \mathbb{E}[|\mathbb{E}_S|], |\mathbb{E}| + \mathbb{E}[|\mathbb{E}_S|]))$, and the memory complexity is $O(g_{\text{GNN}}(|\mathbb{E}| + \mathbb{E}[|\mathbb{E}_S|], |\mathbb{E}| + \mathbb{E}[|\mathbb{E}_S|]))$.

To summarize our analysis, the overall computational complexity is $O(f_{\text{Aug}}(\mathbb{E}[|\mathbb{V}_S|], |\mathbb{E}[|\mathbb{E}_S|]) + f_{\text{GNN}}(|\mathbb{E}| + \mathbb{E}[|\mathbb{E}_S|], |\mathbb{E}| + \mathbb{E}[|\mathbb{E}_S|]))$, and the overall memory complexity is $O(g_{\text{Aug}}(\mathbb{E}[|\mathbb{V}_S|], |\mathbb{E}[|\mathbb{E}_S|]) + g_{\text{GNN}}(|\mathbb{E}| + \mathbb{E}[|\mathbb{E}_S|], |\mathbb{E}| + \mathbb{E}[|\mathbb{E}_S|]))$.

## 4.3 STRENGTHS OF OUR MULTI-VIEW AUGMENTATION SCHEME

### 4.3.1 MULTI-VIEW ON SUBGRAPH

**Augmentation on whole-graph vs. Augmentation on subgraph** The augmentation strategies are supposed to directly modify the whole-graph edges and nodes. However, the time complexity becomes unaffordable for the augmentation strategy when it is greater than $O(|\mathbb{V}|)$. Because the nodes of the subgraphs only accept messages from their $k$-hop neighbors, it's more efficient to augment subgraph-related edges.

**Single-view vs. Multi-view** DropEdge (Rong et al., 2020) and GAug-M (Zhao et al., 2021) take the strategy of single-view augmentation. One way to augment the subgraph-based data is to augment the original graph using the single-view augmentation on subgraph edges to perturb the message-passing flow of the subgraph, which will reduce the amount of the information contained in the original graph. Our multi-view method, on the other hand, can keep the original subgraph structures because it just modifies the augmented subgraph.

**Copy graph vs. Subgraph multi-view** GraphCL (You et al., 2020) creates another graph view to allow contrastive samples to perform the self-supervised training. It's intuitive to create another whole graph by using the augmentation strategies. Nevertheless, such an augmentation flow doubles the memory and time consumptions, which is not affordable for large-scale graph learning tasks like social networks or protein-protein interactions. In this way, multi-view graph learning improves the hardware resource requirements and accelerates the computations.

**Augmentation on all views vs. Preserving one original view** Although augmentation on all views will provide wider training support than keeping the original subgraph structures, however, there is a loss of the information from the original subgraphs. Because we perform the augmentation on the newly-created view, it's enough for perturbing so that keeping one original view can help the model to learn lossless graph-based information. In this way, our proposed multi-view augmentation can show more generalizability and efficiency in subgraph representation learning.

### 4.3.2 STRAIGHT INFERENCE VS. JOINING IN CONTRASTIVE LOSS

Comparing with straight inference, a contrastive loss can effectively make the source subgraphs and the augmented subgraphs closer and the other augmented subgraphs more distant. This strategy will prevent the increased disagreement of views after epochs of training.

## 5 EXPERIMENTS

In this section, we evaluate the performance of our proposed subgraph multi-view augmentation strategy across a variety of architectures, datasets, and graph augmentation strategies, and compare them with other strategies to show the effectiveness of our multi-view subgraph augmentation.

### 5.1 EXPERIMENT SETTINGS

#### 5.1.1 DATASETS

Table 1 summarizes statistics of the datasets obtained from Sub-GNN (Alsentzer et al., 2020). We follow the split reported in Wang & Zhang (2021). Specifically, PPI-BP (Zitnik et al., 2018) aims to pre-

Table 1: Statistics of four real-world datasets.

| DATASET | NODES | EDGES | SUBGRAPHS |
|---------|-------|-------|-----------|
| PPI-BP | 17,080 | 316,951 | 1,591 |
| HPO-METAB | 14,587 | 3,238,174 | 2,400 |
| HPO-NEURO | 14,587 | 3,238,174 | 4,000 |

dict the collective cellular function of a given set of genes known to be associated with specific BP (Biological Processes) in common. The graph shows the correlation of the human PPI (Protein–Protein Interaction) network where nodes represent proteins and edges represent the interaction between proteins. A subgraph is defined by the collaboration of proteins and labeled according to cellular functions from six categories (metabolism, development, signal transduction, stress/death, cell organization, and transport). HPO-METAB and HPO-NEURO (Splinter et al., 2018; Hartley et al., 2020) simulate rare disease diagnosis with the task of predicting subcategories of metabolic and neurological disorders that are the most consistent with these phenotypes. The graph is a knowledge graph containing phenotypic and genotypic information for rare diseases. A subgraph consists of a collection of phenotypes associated with rare monogenic diseases.

#### 5.1.2 GNN MODELS

The proposed augmentation technique is model-agnostic because it does not alter the GNN model. In the experimental part, we evaluate the proposed multi-view augmentation using a subgraph-specific model GLASS (Wang & Zhang, 2021) and two widely-used GNN architectures: GCN (Graph Convolutional Networks) (Kipf & Welling, 2017) and GSAGE (GraphSAGE) (Hamilton et al., 2017). We chose DropEdge (Rong et al., 2020) and GAug-M (Zhao et al., 2021) to create subgraph views that are different from the orignal subgraph.

#### 5.1.3 BASELINES FOR COMPARISON

We compare our proposed approach **w/ Augmentation MV** with the following baselines.

- **Original**: It directly trains the GNN model (i.e., GLASS, GCN, and GSAGE) to establish a baseline without any augmentation (see Figure 1(a)).
- **w/ Augmentation**: It applies DropEdge or GAug-M on the original graph $\mathcal{G}$ to get an augmented graph $\mathcal{G}'$ (see Figure 1(b)). The GNNs will learn on the augmented graph $\mathcal{G}'$.

- **w/ Augmentation Copy**: It applies DropEdge or GAug-M on the original graph $\mathcal{G}$ to get an augmented graph $\mathcal{G}'$ (see Figure 1(c)). The GNNs will learn on both $\mathcal{G}$ and $\mathcal{G}'$. The final embedding is generated after a pooling operation as in Equation 4.

- **w/ Augmentation AllView**: It applies DropEdge or GAug-M on both original subgraphs and the augmented subgraphs, to generate $\mathcal{G}'$ and $\mathcal{G}''$, which fails to preserve the subgraph structure (see Figure 1(d)). The final embedding is generated after a pooling operation as in Equation 4.

### 5.1.4 IMPLEMENTATION DETAILS

We perform a hyperparameter search for all of the baselines and our approach and report the best mean F1 score on all test datasets. The searching space of hyperparameters and details are provided in in the Appendix A. Because GAug-M (Zhao et al., 2021) requires an initial edge probablistic distribution, we use VGAE (Kipf & Welling, 2016) to perform the self-supervised learning on edge prediction to generate the distribution. For the GNN model GLASS (Wang & Zhang, 2021), we first train the model in an unsupervised manner, and then use supervision from downstream tasks to fine-tune the model hyperparameters, following the procedure provided by the original paper. We perform 10 independent training and validation processes with 10 distinct random seeds.

## 5.2 RESULTS

### 5.2.1 OVERALL RESULTS

The empirical performance is summarized in Tables 2 and 3. Our proposed subgraph augmentation improves most of the task accuracy and improves across all three datasets where the whole-graph augmentation strategy succeeded in improving the accuracy, we applied our strategy into GCN, GSAGE, and GLASS. It performs better than most baseline approaches, mainly because it inhibits over-smoothing and over-fitting problems. Specifically, our approach improves the Micro-F1 scores by 0.6%–2.9%, 0.5%–2.5%, and 0.3%–1.7% compared to plain GLASS, GCN, and GSAGE, respectively. Specifically, when we apply multi-view augmentation (MV) to GCN and GSAGE, we find that such backbones combined with multi-view augmentation show a great increase in performance.

Table 2: Mean Micro-F1 scores with standard deviations of the mean on three real-world datasets. Results are provided from runs with 10 random seeds on DropEdge.

| BACKBONE | METHOD | PPI-BP | HPO-METAB | HPO-NEURO |
|----------|--------|--------|-----------|-----------|
| GLASS | Original | $0.610 \pm 0.006$ | $0.600 \pm 0.003$ | $0.678 \pm 0.004$ |
| | w/ DropEdge | $0.626 \pm 0.006$ | $\mathbf{0.607} \pm 0.008$ | $0.678 \pm 0.004$ |
| | w/ DropEdge copy | $0.605 \pm 0.006$ | $0.593 \pm 0.013$ | $0.676 \pm 0.003$ |
| | w/ DropEdge AllView | $0.613 \pm 0.007$ | $0.577 \pm 0.008$ | $0.672 \pm 0.006$ |
| | w/ DropEdge MV | $\mathbf{0.628} \pm 0.005$ | $\mathbf{0.607} \pm 0.008$ | $\mathbf{0.685} \pm 0.003$ |
| GCN | Original | $0.613 \pm 0.008$ | $0.553 \pm 0.018$ | $0.658 \pm 0.007$ |
| | w/ DropEdge | $0.618 \pm 0.006$ | $0.556 \pm 0.006$ | $0.651 \pm 0.006$ |
| | w/ DropEdge copy | $0.553 \pm 0.005$ | $0.349 \pm 0.026$ | $0.317 \pm 0.018$ |
| | w/ DropEdge AllView | $0.595 \pm 0.006$ | $0.552 \pm 0.019$ | $0.613 \pm 0.010$ |
| | w/ DropEdge MV | $\mathbf{0.619} \pm 0.006$ | $\mathbf{0.567} \pm 0.006$ | $0.622 \pm 0.007$ |
| GSAGE | Original | $0.621 \pm 0.006$ | $0.581 \pm 0.008$ | $0.684 \pm 0.002$ |
| | w/ DropEdge | $0.618 \pm 0.006$ | $0.556 \pm 0.006$ | $0.651 \pm 0.006$ |
| | w/ DropEdge copy | $0.593 \pm 0.010$ | $0.555 \pm 0.011$ | $0.676 \pm 0.005$ |
| | w/ DropEdge AllView | $0.618 \pm 0.006$ | $0.567 \pm 0.011$ | $0.682 \pm 0.002$ |
| | w/ DropEdge MV | $\mathbf{0.623} \pm 0.003$ | $\mathbf{0.591} \pm 0.006$ | $\mathbf{0.687} \pm 0.001$ |

**On augmentation on subgraphs** The augmentation should take place in the $k$-hop neighbors of a subgraph to avoid the requirement of extreme fine tunes on hyperparameters, as demonstrated from Tables 2 and 3. Comparing fields of **w/ Augmentation** with **w/ Augmentation MV** we can see that the performance of our approach at least maintains the performance of implementing augmentation on whole graph. On the GSAGE backbone, we can see that DropEdge and GAug-M promote the performance of HPO-NEURO by 3.6% and 2.7%, respectively, compared with the

Table 3: Mean Micro-F1 scores with standard deviations of the mean on three real-world datasets. Results are provided from runs with 10 random seeds on GAug-M.

| BACKBONE | METHOD | PPI-BP | HPO-METAB | HPO-NEURO |
|---|---|---|---|---|
| GLASS | Original | $0.610 \pm 0.006$ | $0.600 \pm 0.003$ | $0.678 \pm 0.004$ |
| | w/ GAug-M | $0.609 \pm 0.003$ | $0.596 \pm 0.007$ | $0.681 \pm 0.003$ |
| | w/ GAug-M copy | $0.621 \pm 0.006$ | $0.593 \pm 0.016$ | $0.679 \pm 0.003$ |
| | w/ GAug-M AllView | $0.617 \pm 0.004$ | $0.579 \pm 0.005$ | $0.672 \pm 0.006$ |
| | w/ GAug-M MV | $\mathbf{0.625} \pm 0.004$ | $\mathbf{0.598} \pm 0.005$ | $\mathbf{0.682} \pm 0.003$ |
| GCN | Original | $0.613 \pm 0.008$ | $0.553 \pm 0.018$ | $0.658 \pm 0.007$ |
| | w/ GAug-M | $0.603 \pm 0.008$ | $0.557 \pm 0.011$ | $0.641 \pm 0.006$ |
| | w/ GAug-M copy | $0.556 \pm 0.007$ | $0.335 \pm 0.025$ | $0.306 \pm 0.016$ |
| | w/ GAug-M AllView | $0.591 \pm 0.006$ | $0.544 \pm 0.008$ | $0.593 \pm 0.007$ |
| | w/ GAug-M MV | $\mathbf{0.616} \pm 0.008$ | $\mathbf{0.564} \pm 0.008$ | $0.640 \pm 0.003$ |
| GSAGE | Original | $0.621 \pm 0.006$ | $0.581 \pm 0.008$ | $0.684 \pm 0.002$ |
| | w/ GAug-M | $0.603 \pm 0.017$ | $0.588 \pm 0.012$ | $0.672 \pm 0.004$ |
| | w/ GAug-M copy | $0.606 \pm 0.006$ | $0.546 \pm 0.007$ | $0.684 \pm 0.003$ |
| | w/ GAug-M AllView | $0.622 \pm 0.005$ | $0.583 \pm 0.001$ | $0.682 \pm 0.003$ |
| | w/ GAug-M MV | $\mathbf{0.628} \pm 0.005$ | $\mathbf{0.591} \pm 0.008$ | $\mathbf{0.689} \pm 0.003$ |

direct augmentation. This result shows that our approach is able to get stronger results within a limited hyperparameter space because we perform the augmentation on neighbor regions, which is a more important region for augmentation.

**On creating subgraph multi-view only** As discussed in Section 4.3.1, one way to perform augmentation is to duplicate a graph. From Tables 2 and 3, comparing fields of **w/ Augmentation copy** with **w/ Augmentation MV**, our approach always performs better than copying the original graph. We even observe a sharp decrease by over 21.8% on the GCN backbone with HPO datasets. This result indicates the importance of the addition of the contrastive loss, and proves that duplicating the original graph may take potential failures in aligning the original graph and the augmented graph.

**On preserving one view** The argument that augmenting all views can bring more training support to models but lose the information from the original graph, as discussed in Section 4.3.1 can also be provided by Tables 2 and 3. Comparing fields of **w/ Augmentation AllView** with **w/ Augmentation MV**, we can find that our approach also brings some improvements. These results echo that it's necessary to keep at least one view to make the subgraphs lossless in information. **w/ Augmentation copy** with **w/ Augmentation MV** field,

### 5.2.2 Ablation Study on the Contrastive Loss

In addition to the comparison on different augmentation strategies, we also perform the ablation on the importance of the contrastive loss, as reported in Tables 4 and 5. Overall, the results show that the contrastive loss can greatly improve the learning process of subgraph embeddings, which utilizes maximizing the agreement between the original subgraph and the augmented subgraph. Also, the decrease on **w/ Augmentation copy** compared with our approach echoes this point of view.

### 5.2.3 Computational Time

We compare the computational time for the direct augmentation on original graphs **w/ Augmentation**, copying the original graph **w/ Augmentation Copy** and our approach **w/ Augmentation MV**, based on the same experimental settings and hyperparameters. We generate the records using GAug-M as the augmentation approach, GLASS as the backbone, and

Table 6: The training time records generated by the experiments.

| METHOD | TIME / EPOCH (s) |
|---|---|
| w/ GAug-M | 0.439 |
| **w/ GAug-M MV** | **0.465** |
| w/ GAug-M copy | 0.788 |

PPI-BP as the dataset. The results are shown in Table 6. We can see that using multi-view does not greatly increase the computational complexity and the memory complexity in practice. Meanwhile, directly duplicating the graph will almost double the training time, which means an unaffordable

Table 4: Ablation Study on DropEdge.

| BACKBONE | METHOD | PPI-BP | HPO-METAB | HPO-NEURO |
|---|---|---|---|---|
| GLASS | Original | $0.610 \pm 0.006$ | $0.600 \pm 0.003$ | $0.678 \pm 0.004$ |
| | w/ No Contrast | $0.618 \pm 0.006$ | $0.597 \pm 0.005$ | $\mathbf{0.685} \pm 0.003$ |
| | w/ Contrast | $\mathbf{0.628} \pm 0.005$ | $\mathbf{0.607} \pm 0.008$ | $0.683 \pm 0.004$ |
| GCN | Original | $0.613 \pm 0.008$ | $0.553 \pm 0.018$ | $0.658 \pm 0.007$ |
| | w/ No Contrast | $0.616 \pm 0.006$ | $0.506 \pm 0.030$ | $0.606 \pm 0.011$ |
| | w/ Contrast | $\mathbf{0.619} \pm 0.006$ | $\mathbf{0.567} \pm 0.006$ | $\mathbf{0.622} \pm 0.007$ |
| GSAGE | Original | $0.621 \pm 0.006$ | $0.581 \pm 0.008$ | $0.684 \pm 0.002$ |
| | w/ No Contrast | $0.616 \pm 0.007$ | $0.583 \pm 0.008$ | $0.684 \pm 0.004$ |
| | w/ Contrast | $\mathbf{0.623} \pm 0.003$ | $\mathbf{0.591} \pm 0.006$ | $\mathbf{0.687} \pm 0.001$ |

Table 5: Ablation Study on GAug-M.

| BACKBONE | METHOD | PPI-BP | HPO-METAB | HPO-NEURO |
|---|---|---|---|---|
| GLASS | Original | $0.610 \pm 0.006$ | $0.600 \pm 0.003$ | $0.678 \pm 0.004$ |
| | w/ No Contrast | $0.611 \pm 0.006$ | $0.591 \pm 0.007$ | $\mathbf{0.682} \pm 0.003$ |
| | w/ Contrast | $\mathbf{0.625} \pm 0.004$ | $\mathbf{0.598} \pm 0.005$ | $0.680 \pm 0.003$ |
| GCN | Original | $0.613 \pm 0.008$ | $0.553 \pm 0.018$ | $0.658 \pm 0.007$ |
| | w/ No Contrast | $0.613 \pm 0.008$ | $0.513 \pm 0.019$ | $0.580 \pm 0.015$ |
| | w/ Contrast | $\mathbf{0.616} \pm 0.008$ | $\mathbf{0.564} \pm 0.008$ | $\mathbf{0.640} \pm 0.003$ |
| GSAGE | Original | $0.621 \pm 0.006$ | $0.581 \pm 0.008$ | $0.684 \pm 0.002$ |
| | w/ No Contrast | $0.621 \pm 0.005$ | $0.582 \pm 0.008$ | $0.684 \pm 0.003$ |
| | w/ Contrast | $\mathbf{0.628} \pm 0.005$ | $\mathbf{0.591} \pm 0.008$ | $\mathbf{0.689} \pm 0.003$ |

increase on the computational complexity. This result shows that our approach gains both efficiency and effectiveness in subgraph augmentation comparing with other baseline approaches.

## 6 CONCLUSION

We propose a novel model-agnostic subgraph augmentation strategy to facilitate subgraph-based GNNs. By creating a new subgraph and link to the original graph, it will include more diversity in message passing to the graph to enhance model training support. This subgraph-specific augmentation strategy can improve the performance and the robustness of a graph neural network. A bunch of improved experiments on both GAug-M and DropEdge on different datasets show the generalizability on models and augmentation strategies. It's expected that our research will empower the subgraph representation learning to go further and broader.

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

## A APPENDIX

### A.1 IMPLEMENTATION DETAILS

#### A.1.1 HYPERPARAMETER TUNING

By solidating the hyperparameters by GLASS on all backbone settings, we implement the grid hyperparameter searching on augmentations. For $\lambda$, we select a searching space $\{0, 0.01, 0.05, 0.1, 0.5, 1, 1.25, 1.5, 2, 2.5\}$. For DropEdge-related tasks, our searching space for $p$ is $\{0.1, 0.2, 0.3, 0.4, 0.5\}$. For GAug-related tasks, our searching space for (rm_pct, add_pct) is $\{(0, 50), (15, 35), (30, 20), (45, 5)\}$. Since we used VGAE to generate different probabilistic edge distribution, we selected $i$ from $\{1, 2\}$.

