# OpenReview forum: "Improving Subgraph Representation Learning via Multi-View Augmentation"
_ICLR.cc/2023/Conference — Submitted to ICLR 2023_

### Official Review · Reviewer_mpjc · 2022-10-22

**Confidence:** 3
**Correctness:** 3
**Technical Novelty And Significance:** 2
**Empirical Novelty And Significance:** Not applicable
**Recommendation:** 3

**Clarity, Quality, Novelty And Reproducibility:**

The paper is clearly written, but its technical contribution is limited. The proposed multi-view strategy works better than basic GNNs but further experimental evaluations are needed.

**Strength And Weaknesses:**

Strengths:
1. The paper is clearly written, and the proposed framework is well articulated.
2. The idea of this paper is straightforward and easy to understand.
3. Experimental evaluation shows that the multi-view strategy works well when collaborating with two graph augmentation methods on 3 GNN models.

Weaknesses:
1. The technical contribution of the paper is limited.
The main idea behind the proposed multi-view strategy is to generate augmented samples by adding perturbations to the original (sub)graph, and to maximize the similarity between the representations of these (sub)graphs. However, similar ideas have been around in the literature of contrastive learning. Although most graph contrastive learning methods are designed for node- or graph-level tasks, they can be easily applied to subgraphs [1] [2].

2. The application potential of the proposed method is limited.
The paper only adopts the proposed multi-view strategy to subgraph tasks, which could limit its application potential. A natural question is: can this multi-view strategy also improve node- and/or graph-level tasks (e.g., when combined with graph pooling methods [3] [4]).

3. Some implementation details are missing, which could harm the reproducibility.
   (1) For the pooling function Pool() in Eq. (4), which one was chosen in the experiments? Is there any option other than MaxPool and AvgPool?
   (2) What is the design of the contrastive loss in Eq. (6)?

4. Experimental study should be enhanced.
   (1) The authors only conduct experiments with basic GNNs, such as GLASS, GCN and GraphSAGE. More SOTA GNN methods should be added to the comparison as well.
   (2) No comparison with Graph Contrastive Learning methods was performed.
   (3) The choice of the hyperparameter λ should be further explored.

5. Some typos.
In the caption of Figure 1, blue -> green, G" -> G’’


[1] Suresh S, Li P, Hao C, et al. Adversarial graph augmentation to improve graph contrastive learning. Advances in Neural Information Processing Systems, 2021, 34: 15920-15933.
[2] Yin Y, Wang Q, Huang S, et al. Autogcl: Automated graph contrastive learning via learnable view generators. Proceedings of the AAAI Conference on Artificial Intelligence. 2022, 36(8): 8892-8900.
[3] Lee J, Lee I, Kang J. Self-attention graph pooling. International conference on machine learning. PMLR, 2019: 3734-3743.
[4] Zhang Z, Bu J, Ester M, et al. Hierarchical graph pooling with structure learning. arXiv preprint arXiv:1911.05954, 2019.


**Summary Of The Paper:**

The paper proposes a multi-view strategy for subgraph representation learning. The multi-view strategy aims to achieve better performance by combining with arbitrary graph data augmentation methods and GNN models.  Experimental results are provided to demonstrate the effectiveness of the proposed strategy.

**Summary Of The Review:**

Major concerns are: (1) the technical contribution of the proposed strategy is limited; (2) Baseline models are weak. More comparisons are needed. (3) Some implementation details are missing.

---

### Official Review · Reviewer_7uEX · 2022-10-25

**Confidence:** 3
**Correctness:** 3
**Technical Novelty And Significance:** 2
**Empirical Novelty And Significance:** 2
**Recommendation:** 5

**Clarity, Quality, Novelty And Reproducibility:**

The paper is well-written and easy to read. Well executed and of good quality. The novelty is a bit low.

**Strength And Weaknesses:**

The paper is easy to follow and well-presented, with a good set of well-conducted experiments and ablation studies.

1) My biggest concern is the technical depth of this work. It would be helpful if the author would further characterize the augmentations and demonstrate to us how multiple-view + augmentation affects subgraph learning. And what exact benefits would we get? Under what conditions?
2) Related to 1), it would be helpful if we can see how this method work on a larger set of various tasks.
3) As for the performance, a question is how significantly augmentation helps. It looks like in most cases, the margin between the original and the augmented multiple views is small and likely not statistically significant. It seems to help more in some of the GCN cases.
4) For computational time comparison in table 6, please add the original, i.e., w/o augmentation


**Summary Of The Paper:**

The authors proposed a multi-view approach applied to both the original and augmented subgraphs to learn subgraph embeddings/tasks. They used existing graph augmentation techniques, and then align the augmented new graph with the original one and extract subgraph embeddings using available subgraph GNNs.

**Summary Of The Review:**

A well-executed, good quality paper, yet I would like to see a bit more depth and novelty.

---

### Official Review · Reviewer_qUyv · 2022-10-26

**Confidence:** 4
**Correctness:** 2
**Technical Novelty And Significance:** 2
**Empirical Novelty And Significance:** 2
**Recommendation:** 3

**Clarity, Quality, Novelty And Reproducibility:**

**************Clarity**************

The paper is clearly written to understand overall method.

**************Quality**************

The performance gain is marginal compared to existing methods.

**********Novelty**********

The proposed method has a limited novelty.

******************************Reproducibility******************************

The authors share their source code in the supplement. The paper has good reproducibility.

**Strength And Weaknesses:**

**Strengths**

(1) This paper greatly improves the efficiency of graph augmentation for subgraphs.

**Weakness**

(1) I think that the novelty of this paper is limited. To augment subgraphs, this paper uses existing graph augmentation methods such as DropEdge and GauG. Also, the contrastive loss of proposed method is from MV-GNN. Even though embedding augmented subgraphs into original graph is somewhat novel, I think it would be better to propose subgraph specific graph augmentation methods.

(2) The proposed methods have marginal performance gain compared to baselines. For example On PPI-BP, GLASS w/ DropEDGE MV has 0.002 performance gain compared to GLASS w/ DropEDGE.

**Summary Of The Paper:**

The paper proposes multi-view augmentation for learning representations on subgraphs. The proposed method improves the efficiency of existing standard graph augmentation method.

**Summary Of The Review:**

Overall, I am leaning towards rejection. My major concern is the novelty and performance of proposed methods. If you address my concerns, I will raise my score.

---

### Decision · Program_Chairs · 2023-01-20

**Decision:**

Reject

**Justification For Why Not Higher Score:**

The authors did not submit any author response.

**Justification For Why Not Lower Score:**

N/A

**Metareview: Summary, Strengths And Weaknesses:**

This paper proposes a mutli-view augmentation method for subgraph representation learning, where the property of a subgraph within a whole graph is to be predicted. The method improves the efficiency, and is clearly presented. However, all reviewers agree that the novelty and technical depth of the method are limited, and the performance improvement is marginal.